# Studies on the Accumulation of Secondary Metabolites and Evaluation of Biological Activity of In Vitro Cultures of *Ruta montana* L. in Temporary Immersion Bioreactors

**DOI:** 10.3390/ijms24087045

**Published:** 2023-04-11

**Authors:** Agnieszka Szewczyk, Andreana Marino, Maria Fernanda Taviano, Lucia Cambria, Federica Davì, Monika Trepa, Mariusz Grabowski, Natalizia Miceli

**Affiliations:** 1Department of Pharmaceutical Botany, Faculty of Pharmacy, Jagiellonian University Medical College, 30-688 Krakow, Poland; 2Department of Chemical, Biological, Pharmaceutical and Environmental Sciences, University of Messina, Viale F. Stagno d’Alcontres, 31, 98166 Messina, Italy; 3Foundation “Prof. Antonio Imbesi”, University of Messina, Piazza Pugliatti 1, 98122 Messina, Italy

**Keywords:** in vitro cultures, *Ruta montana*, coumarins, alkaloids, phenolic compounds, antibiofilm formation, antibacterial activity, antioxidant activity

## Abstract

The present work focuses on in vitro cultures of *Ruta montana* L. in temporary immersion Plantform^TM^ bioreactors. The main aim of the study was to evaluate the effects of cultivation time (5 and 6 weeks) and different concentrations (0.1–1.0 mg/L) of plant growth and development regulators (NAA and BAP) on the increase in biomass and the accumulation of secondary metabolites. Consequently, the antioxidant, antibacterial, and antibiofilm potentials of methanol extracts obtained from the in vitro-cultured biomass of *R. montana* were evaluated. High-performance liquid chromatography analysis was performed to characterize furanocoumarins, furoquinoline alkaloids, phenolic acids, and catechins. The major secondary metabolites in *R. montana* cultures were coumarins (maximum total content of 1824.3 mg/100 g DM), and the dominant compounds among them were xanthotoxin and bergapten. The maximum content of alkaloids was 561.7 mg/100 g DM. Concerning the antioxidant activity, the extract obtained from the biomass grown on the 0.1/0.1 LS medium variant, with an IC_50_ 0.90 ± 0.03 mg/mL, showed the best chelating ability among the extracts, while the 0.1/0.1 and 0.5/1.0 LS media variants showed the best antibacterial (MIC range 125–500 µg/mL) and antibiofilm activity against resistant *Staphylococcus aureus* strains.

## 1. Introduction

Plants of the *Ruta* genus have a rich and varied chemical composition. They mainly contain compounds from the group of coumarins, alkaloids, flavonoids, phenolic acids, and essential oil [1,2]. *Ruta montana* is widespread in the Mediterranean region since it grows in sandy and dry areas, specifically in countries such as Algeria, Morocco, and Tunisia. These plants also grow in Greece, Portugal, and Turkey in Europe [3,4,5,6]. The diverse chemical composition of the species includes compounds from groups of alkaloids, flavonoids (flavanols, including catechins and flavones), coumarins (hydroxycoumarins and furanocoumarins), organic acids, tannins, among other essential oils (sesquiterpenes and monoterpenes). The genus *R. montana* also contains derivatives of leucoanthocyanins, sterols, triterpenes, mucus, monosaccharides, and anthracene [1,2,3,7,8,9,10]. Additionally, the diverse chemical composition of *R. montana* favors multidirectional therapeutic uses. The species further contains furanocoumarins, such as bergapten and xanthotoxin, which are used in dermatology to treat vitiligo and psoriasis [11,12]. Although these compounds have a therapeutic effect, they also have some toxic effects, such as photodermatitis as well as kidney and liver damage [6]. Therefore, their application should be controlled. The raw materials of *R. montana* have antidiabetic properties, which cause a decrease in the blood glucose concentration and improve glucose tolerance. Further, they may also have a positive effect on the architecture of the pancreatic islets [2,13]. The extracts of *R. montana* also show antioxidant activity due to the high content of phenolic compounds. The compounds have the ability to scavenge free radicals [2,6]. The essential oil obtained from *R. montana* has antifungal, insecticidal, and larvicidal properties, which showed an inhibitory effect on the growth of the *Candida albicans* and *Aspergillus niger* species [2,3]. Additionally, the species have proven antimicrobial properties. Rue extracts from the species inhibited the growth of pathogens, such as *Staphylococcus aureus*, *Bacillus subtilis*, *Listeria innocua*, *Proteus mirabilis*, and *Clavibacter michiganensis*. The ketone components of the essential oil are the substances that are responsible for this activity [7]. The extracts from *R. montana* are also considered to be a potential antihypertensive drug. Studies conducted on rats suffering from hypertension revealed that this agent lowered systolic and diastolic blood pressure, mean blood pressure, and heart rate. Moreover, it caused dose-dependent relaxation of the aorta. It is likely that the prostaglandin pathway mediates these processes [14].

In vitro plant cultures have great potential for the commercial production of secondary metabolites. This method has many advantages. It allows the synthesis of bioactive compounds under controlled conditions, regardless of climate and soil conditions. In vitro plant cultures can be carried out continuously using selected cell lines, which makes them a valuable and reliable source of secondary metabolites [15].

A modern method of cultivation, a temporary immersion system (Plantform^TM^), was selected for the in vitro cultures. This type of bioreactor allows for obtaining a large amount of biomass in a short period of time [16]. Plantform^TM^ bioreactors have recently been used to produce secondary metabolites. They are very useful for cultivating shoot cultures, the enlargement of which on a larger scale has been a major technical problem so far. This method allowed the production of secondary metabolites from various groups of compounds on a larger scale. The authors of studies on *Schizandra chinensis* cultures conducted in various types of bioreactors found the highest production of lignans in cultures carried out in Plantform^TM^ bioreactors [17]. Subsequently, *Centella asiatica* cultures in Plantform^TM^ bioreactors turned out to be a valuable source for obtaining asiaticoside, phenolic acids, and flavonoids [18]. Cultures of *Scutellaria lateriflora* maintained in this type of bioreactor showed a high ability to accumulate verbascoside and flavonoids [19]. The in vitro plant cultures are considered an alternative high-quality plant material for pharmaceutical or cosmetic purposes. Currently, there are no studies on the in vitro cultures of *R. montana*. Therefore, the present study aimed to investigate the biosynthetic, antioxidant, antibacterial, and antibiofilm potential of the in vitro cultures of *R. montana* using a temporary immersion system (Plantform^TM^).

## 2. Results and Discussion

### 2.1. Biomass Growth

The *R. montana* cultures grown in bioreactors mainly in the form of shoots (Figure 1). The cultures were characterized by an 8-fold increase in biomass. The content of dry biomass obtained from the *R. montana* bioreactor cultures ranged between 8.042 (LS medium containing 0.5/1.0 mg/L NAA/BAP, 5-week growth cycle) and 7.247 g (LS medium containing 0.1/0.1 mg/L NAA/BAP, 6-week growth cycle). The maximum weight was obtained after a 5-week cultivation period (growth cycle) on the LS medium containing 0.5/1.0 mg/L NAA/BAP. There were no significant differences in biomass growth after 5 and 6 weeks of the culture cycle for most of the LS medium. Only in case of the LS 0.5/0.5 medium variant was the dry biomass after 5 weeks significantly higher compared to those for 6 weeks. This was due to the gradual withering away of cultures. Macroscopic observations of the tissue showed the cultures had a tendency to die out after 6 weeks. An intense green color was observed on the 5-week-old cultures, while the 6-week-old cultures were partially covered in brown. The content of obtained dry biomass is summarized in Table 1.

The increase in dry biomass varied from 8.9-fold (LS 0.1/0.1, 6 weeks) to 9.9-fold (LS 0.5/1.0, 5 weeks). Previous studies on the increase in biomass in in vitro cultures of another species of the *Ruta* genus indicated a wide variation, depending on the type of culture used, the duration of its cultivation, and the medium variant used. In the studies on the accumulation of furanocoumarins in *R. graveolens* shoot cultures, which were conducted for 6 weeks in liquid stationary cultures on medium with NAA/BAP 2.0/2.0 mg/L, a 5-fold growth of dry biomass was observed. The highest increase occurred between the 7th and 21st day of culture [20]. In the next studies, the in vitro cultures of *Ruta graveolens* ssp. *divaricata* were grown for 4 weeks on the LS medium, supplemented with NAA/BAP at various concentrations, ranging between 0.1 and 3.0 mg/L. The fold increase in the dry biomass was 6.2–6.3 [21]. Another research was carried out on agitated cultures on the species *Ruta corsica*, *Ruta chalepensis*, and *Ruta graveolens*, on the LS medium with the addition of NAA/BAP at a concentration of 0.1/0.1 mg/L for a period of 3, 4, 5, 6, and 7 weeks. The highest increase in biomass for *R. graveolens* was observed for the culture grown for 4 weeks (33.3-fold increase), while for *R. chalepensis* and *R. corsica*, the highest increase occurred after 5 weeks (33.7-fold and 31.3-fold increase, respectively) [22]. The increase in biomass in cultures of other plant species cultivated in temporary immersion bioreactors differ depending on the duration of the culture and the media variants used. The microshoot cultures of *Scutellaria lateriflora* were maintained in Plantform^TM^ bioreactors on two media (LS and MS—Murashige—Skoog, both contained 0.5/1.0 mg/L NAA/BAP) for 4 weeks. The increase in dry biomass was 6.3-fold on MS medium and 6.9-fold on the LS medium [19]. Shoot cultures of *Schizandra chinensis* conducted in Plantform^TM^ bioreactors on MS medium containing 1.0/3.0 mg/L NAA/BAP showed good increase in biomass (from 2.5-fold after 30 days of cultivation to 9-fold after 60 days of cultivation) [17].

In conclusion the *R. montana* cultures showed good biomass growth. There are no statistically significant differences between the media used.

### 2.2. Phytochemical Investigation

#### 2.2.1. HPLC Analysis

The HPLC analysis of methanol extracts obtained from the biomass of *R. montana* bioreactor cultures identified various secondary metabolites related to alkaloids and phenolic compounds (coumarins and catechins) (Table 2). There were only small amounts of phenolic acids accumulated. Spectrophotometric determinations indicated the presence of polyphenolic and flavonoid compounds in the extracts (Table 3). However, according to the HPLC analysis, there were no flavonoids corresponding to the reference substances used. The absence of the main flavonoid—rutoside, which is typically present in the parent plant, was particularly significant. The absence can be explained due to the fact that in vitro cultures generally tend to have different metabolic pathways compared to the parent plants. Some of the enzymes can also become inactive or complex glycosidic bonds, which are not present in the parent plant, often form due to the glycosylation process [15]. The HPLC analysis confirmed the accumulation of linear furanocoumarins (bergapten retention time (RT) = 63.72 min, isoimperatorin RT = 70.55 min, isopimpinellin RT = 60.91 min, psoralen RT = 53.38 min, xanthotoxin RT = 54.18 min), furoquinoline alkaloids (γ-fagarine RT = 64.68 min, 7-isopentenyloxy-γ-fagarine RT = 71.92 min, skimmianine RT = 62.36 min), and catechins (catechin RT = 7.3 min). Sample chromatograms of the extract from *R. montana* in vitro cultures are included in Appendix A (Appendix A).

The dynamics of the accumulation of individual metabolites varied according to the time of culture and the LS medium variant. Overall, the production of the metabolites was higher after the 5-week growth cycle. The exceptions are isopimpinellin and 7-isopentenyloxy-γ-fagarine, which, on most variants of the LS medium, reached a higher content after a 6-week growth cycle. Table 2 shows the mean contents [mg/100 g DW] of individual metabolites depending on the LS medium variant and the growth cycle. A comparison of homogeneous groups marked with the letters a-g was used, where “a” means the group of lowest mean, followed by “b” for the next average, and so on to the highest averages. The results of detailed statistical analyzes and the list of homogeneous groups are included in the Appendix A, for each metabolite separately.

Among the coumarins, the highest accumulated individual metabolite was xanthotoxin (maximum content 885.9 mg/100 g DW, LS 0.5/0.5 medium, 5-week growth cycle). The second highest accumulated metabolite was the coumarin bergapten, with the maximum content obtained after the 5-week growth cycle on the 1.0/1.0 LS medium (446 mg/100 g DW). The quantity of psoralen was also high, with the maximum content of 340.1 mg/100 g DW (5-week growth cycle, LS 0.1/0.1). The amounts of isopimpinellin and isoimperatorin were lower, with the maximum content of 223.2 (6-week growth cycle, LS 1.0/1.0) and 105.8 mg/100 g DM (5-week growth cycle, LS 1.0/1.0), respectively.

Among the confirmed furoquinoline alkaloids, γ-fagarine was accumulated in the highest amount (305.4 mg/100 g DW) on the 0.1/0.1 LS medium, after the 5-week growth cycle. The maximum skimmianine content was 233.7 mg/100 g DW (5-week growth cycle, LS 0.5/1.0). Isopentenyloxy-γ-fagarine was accumulated in the lowest amount, with the maximum content of 42.2 mg/100 g DW (6-week growth cycle, LS 0.5/0.5).

In the analyzed catechins, catechin was accumulated after the 5-week growth cycle, in the maximum amounts of 89.6 mg/100 g DW on the 0.1/0.1 LS medium; this result is in agreement with those obtained with the spectrophotometric determinations of the condensed tannins (Table 2 and Table 3).

All the analyzed extracts had very high furanocoumarin total content. The maximum content of 1824.3 mg/100 g DW was obtained on the 0.1/0.1 LS medium after the 5-week growth cycle. High contents of furanocoumarins were also noted after the 5-week growth cycle on LS 0.5/0.5 and LS 1.0/1.0 media 1657.6 and 1582.9 mg/100 g DW, respectively.

The maximum content of the analyzed furoquinoline alkaloids (561.6 mg/100 g DW) was also obtained on the 0.1/0.1 LS medium after the 5-week growth cycle.

The obtained results suggest that *R. montana* bioreactor cultures can be proposed as an alternative and rich in vitro controlled source of linear furanocoumarins and furoquinoline alkaloids. Xanthotoxin and bergapten have photosensitizing effects on human skin and their pigmentation-stimulating and antiproliferative properties are utilized in the symptomatic treatment of vitiligo, psoriasis, and mycosis fungoides. Therefore, Bergapten is better tolerated by patients [11,12]. The obtained results showed very high levels of xanthotoxin (above 800 mg/100 g DW) and bergapten (above 400 mg/100 g DW). For comparison, the content of these compounds in plants cultivated in the field is 410 and 110 mg/100 g DW, respectively [23]. The biosynthetic potential for the production of furoquinoline alkaloids is also an interesting area of study. Furoquinoline alkaloids show a number of biological activities, including antifungal and antibacterial properties, inhibitory activity toward AchE (acetylcholinesterase), and 5-HT2 receptor-inhibiting properties [24]. The content of the above-mentioned groups of metabolites in cultures of another species of rue—*R. graveolens* varies depending on the type of culture, culture conditions, and the strategies used to increase the production of bioactive metabolites. Study on the level of accumulation of furanocoumarins in stationary liquid shoot cultures of *R. graveolens* confirmed the presence of the following coumarins: psoralen, xanthotoxin, isopimpinellin, bergapten, imperatorin, and umbelliferon. The maximum content of coumarins (966 mg/100 g DW) was determined after a 4-week growth cycle in the LS medium containing 2/2 mg/L NAA/BAP, and the dominant metabolites were found to be xanthotoxin (330 mg/100 g DW) and bergapten (320 mg/100 g DM) [20]. In subsequent studies performed in stationary liquid cultures maintained under various light conditions on the LS medium containing 2/2 mg/L NAA/BAP (6-week breeding cycle), the highest total content of coumarins (1022 mg/100 g DM) was observed in the cultures grown under white constant artificial light. The maximum content of the main furanocoumarins (xanthotoxin and bergapten) was 433.4 and 219.5 mg/100 g DM, respectively [25]. Similar to previous studies, in the study on *R. graveolens* agitated cultures, it was observed that the most dominant coumarins were xanthotoxin (428.3 mg/100 g DW) and bergapten (186.6 mg/100 g DW). The highest total content (917.2 mg/100 g DW) of linear furanocoumarins was reached after a 5-week growth cycle on the LS medium containing 0.1/0.1 mg/L NAA/BAP. The dominant furoquinolic alkaloids were skimmianine (94.6 mg/100 g DW) and γ-fagarine (54.5 mg/100 g DW). The highest total content (155.9 mg/100 g DW) of these alkaloids was noted after a 5-week growth cycle on the LS medium containing 0.1/0.1 mg/L NAA/BAP [22].

Use of abiotic and biotic elicitors in agitated shoot cultures of *R. graveolens* caused an increase in the production of furanocoumarins and furoquinoline alkaloids. The highest obtained contents of the main coumarins were as follows: xanthotoxin (288.36 mg/100 g DW, 8.5-fold increase, compared to control cultures), bergapten (153.78 mg/100 g DW, 3.7-fold increase) The highest obtained contents of main furoquinoline alkaloids were as follows: γ-fagarine (68.0 mg/100 g DW), skimmianine (48.0 mg/100 g DW) [26,27,28].

#### 2.2.2. Total Phenolic, Flavonoid, and Condensed Tannin Content

Total content of phenolic, flavonoid, and condensed tannins was measured in extracts from biomass obtained after 5-week growth cycle. The results of Folin–Ciocâlteu assay indicated that the total phenolic content was highest in the extract of *R. montana* biomass grown on the LS medium variant supplemented with NAA/BAP at the concentration of 0.1/0.5 mg/L, followed by the in vitro culture maintained on the LS medium variant 1.0/1.0 (Table 3).

Similar trend was observed for the total flavonoid content that, in the different extracts, varied from 16.96 to 45.65 mg QE/g extract. The extract obtained from in vitro culture maintained on the LS medium variant 0.1/0.5 contained the highest content of flavonoids, followed by the extract maintained on the LS medium variant 1/1 (Table 3).

The condensed tannin content, as evaluated by the vanillin assay, was found to be low in all the extracts. Among the extracts, that obtained from *R. montana* in vitro culture maintained on the LS medium variant 0.1/0.1 resulted as the richest (10.97 ± 0.50 mg QE/g extract) (Table 3). The content of condensed tannins determined by spectrophotometric method followed the same trend observed for these compounds determined by HPLC analysis; in fact, the content of condensed tannins as well as of catechins detected by HPLC resulted higher in the extract obtained from biomass grown on the variant LS 0.1/0.1 and decreased in the same order (Table 2 and Table 3).

Recently, the content of total polyphenols was determined spectrophotometrically in a methanolic extract obtained from the aerial parts of *R. montana* collected in Algeria and in Morocco [29,30]. Interestingly, our extracts, obtained from biomass cultured in vitro, were found much richer both in phenolic compounds and in flavonoids than those prepared from the same species growing in field, demonstrating that the in vitro plant cultures can be considered an alternative high-quality plant material for pharmaceutical or cosmetic purposes. The content of phenolic compounds in cultures of other species of rue has not been extensively studied. This is due to the generally low content of these compounds. In the study on *R. graveolens* shoots grown in stationary liquid cultures on four different variants of the LS medium (with NAA and BAP added at different concentrations, in the range from 0.1 to 3.0 mg/L), the total content of phenolic acids was determined in the range from 85.04 to 108.28 mg/100 g DW (depending on the medium variant). The highest content of the tested phenolic acids was observed on the LS medium containing 2.0/2.0 mg/L NAA/BAP [31]. In the study on effect of monochromatic light conditions on the production of free phenolic acids in stationary liquid shoot cultures of R. graveolens, the highest total content (103.4 mg/100 g DW) of phenolic acids was observed in the biomass from the cultures cultivated on the LS medium, containing 3.0/1.0 mg/L NAA/BAP under white light [32]. An attempt was also made to increase the production of phenolic compounds in agitated shoot cultures of *R. graveolens* using the phenylalanine feeding method. The addition of this precursor increased 1.5-fold the production of phenolic compounds (phenolic acids and catechins). The highest obtained contents were, respectively, 109 mg/100 g DW (total content of phenolic acids) and 65.9 mg/100 g DW (catechin content) [33].

### 2.3. Antioxidant Activity

Reactive oxygen species (ROS) are widely regarded as etiologic factors for several diseases, including cancer, inflammation, and organ injuries. Evidence suggests that the antioxidants, scavenging ROS in pathological condition, may reduce cell damage and control the pathological process [34]. Antioxidants can be divided into primary (or chain-breaking) and secondary (or preventive) categories, and the primary antioxidant reactions can be classified into hydrogen-atom transfer (HAT) and single-electron transfer (SET). The HAT mechanism takes place when an antioxidant scavenges free radicals donating hydrogen atoms; an antioxidant acting by SET mechanism transfers a single electron to reduce any compound. Antioxidants can act through various mechanisms, and therefore it is vital to use methodologies with distinct mechanisms to evaluate the antioxidant capacity of plant-derived phytocomplexes or isolated compounds.

Based on the different mechanisms of determination of the antioxidant capacity, three in vitro tests were used to determine the in vitro antioxidant effectiveness of *R. montana* extracts: the DPPH assay (involving both HAT and SET mechanisms), and the reducing power assay (based on SET mechanism); the ferrous ions chelating activity was determined to assess the secondary antioxidant ability.

The DPPH test is the most common spectrophotometric method used to evaluate the free scavenging properties of a phytocomplexes or pure compounds due to its simple, rapid, sensitive, and reproducible procedures. Figure 2A shows the results of the DPPH test; it is evident from the comparison of all extracts of the species of *R. montana* and the reference standard BHT that, in the range of concentrations assayed, all of them display a weak activity. This result is confirmed by the IC_50_ values as well, which were higher than 2 mg/mL for all the extracts (Table 3).

The Fe^3+^–Fe^2+^ transformation method was used to evaluate the reducing power of the *R. montana* extracts, which show that all extracts displayed mild activity compared to the standard BHT (Figure 2B). The calculated ASE/mL values indicated that the best reducing efficacy was highlighted for the extracts obtained from *R. montana* biomass grown on 1/1 and 0.5/0.5 LS medium variants (18.28 ± 4.30 and 19.54 ± 0.64 ASE/mL, respectively) (Table 3).

The *R. montana* extracts exhibited good, dose-dependent, and chelating properties in the Fe^2+^ chelating activity assay (Figure 2C). Comparing the IC_50_ values calculated for all the extracts, the values obtained from biomass in vitro cultured on the 0.1/0.1 LS medium variant, with an IC_50_ of 0.90 ± 0.03 mg/mL, with about 84% activity at the maximum concentration assayed, resulted as the most active, followed by those from in vitro culture grown on the 0.5/0.5 and 0.5/1 LS medium variants (Table 3).

It is evident from the obtained results that *R. montana* extracts act as weak primary antioxidants and possess good secondary antioxidant properties.

*Ruta* genus is rich in bioactive phytochemicals, such as coumarins, phenolic acids, flavonoids, alkaloids, and tannins [6]. The phytochemical investigations carried out by HPLC have highlighted a different metabolic profile of the in vitro cultures of *R. montana* than that of the parent plant. The flavonoids in all the extracts were undetectable, and phenolic acids were only present in small amounts. The main secondary metabolites in *R. montana* cultures were linear furanocoumarins (the dominant compound among them was xanthotoxin), followed by furoquinoline alkaloids (γ-fagarine, 7-isopentenyloxy-γ-fagarine, skimmianine) and catechins. It was reported that coumarins imperatorin, xanthotoxin, and bergapten did not possess antiradical activity against DPPH, though they exhibited a moderate level of reducing power and were effective in the ferrous ion-chelation test [35]. Coumarins reduce the level of oxidative stress via chelation of redox-active Cu and Fe, thus suppressing the ROS formation via the Fenton reaction [36]. It was reported that catechin can act as free radical scavengers as well as a metal chelator [37].

It is evident from the result of the chelating activity assay that the extract obtained on the 0.1/0.1 LS medium variant is the most effective; this extract is also the richest in coumarins and catechin. Therefore, the good secondary antioxidant properties highlighted for all *R. montana* extracts could mainly be related to the great amount of coumarins and catechin.

### 2.4. Antibacterial Screening

There has been an increase in the spread of resistant bacterial strains due to the continued inappropriate usage of antibiotics. Many of these produce biofilms, which are a complex bacterial community encased in an extracellular polymeric matrix notoriously difficult to eradicate once established [38]. The biofilm network possesses the ability to evade environmental threats, such as antimicrobials and host defense mechanisms [39]. Due to this global health issue, there has been extensive research on finding alternative therapeutics. Plants rich in natural secondary metabolites are one of the go-to reservoirs in discovering potential resources to alleviate this problem [40]. In this study, the antibacterial screening of *R. montana* extracts was performed against a representative set of Gram-positive and Gram-negative bacterial strains. All the extracts obtained from in vitro culture of *R. montana* grown on the LS medium variants supplemented with NAA/BAP at different concentrations showed selective antibacterial activity. Table 4 shows the results of the antibacterial activity screening of extracts of the *R. montana* biomass grown on the different LS medium variants. The extract the 0.1/0.1 LS medium showed the highest activity against all the *Staphylococcus* strains and *E. coli* ATCC 10536 with the best effect against *S. aureus* ATCC 6538, *S. aureus* ATCC 43300, and *S. aureus* 74CCH (MIC = 125 μg/mL; MBC = 250–500 μg/mL), followed by the 0.5/0.5 LS medium active against *S. aureus* ATCC 43300 (MIC = 125 μg/mL) and *S. epidermidis* ATCC 35984 (MIC = 250 μg/mL), the 0.5/1.0 LS medium active against *S. aureus* 74CCH (MIC = 125 μg/mL) and *S. epidermidis* ATCC 35984 (MIC = 250 μg/mL), and the 0.1/0.5 LS medium active against *S. aureus* ATCC 43300 (MIC = 125 μg/mL). The extracts showed no activity against *E. coli* DSM 105388, *P. aeruginosa* ATCC 9027, and *P. aeruginosa* DSM 102273. The presence of a high content of xanthotoxin, followed by bergapten, psoralen, γ -fagarine, and catechins, likely justifies the efficacy of these extracts [41,42]. The good activity of the 0.5/0.5 LS medium against all the *Staphylococcus* strains can be due to the high content of xanthotoxin, followed by bergapten and skimmianine [41,42,43,44]. Coumarins have the ability to bind to the B subunit of DNA gyrase in bacteria and can inhibit DNA supercoiling by blocking the ATPase activity [45]. Alkaloids, such as γ-fagarine, have been shown to exhibit antimicrobial activity by inhibiting enzyme activity or other mechanisms, such as bacterial membrane disruption of Gram-negative bacteria [43]. Catechins demonstrated bactericidal effects on both Gram-positive and Gram-negative bacteria, including multidrug-resistant strains through membrane disruption and damage in DNA and protein oxidation [46].

In a previous study, the extracts obtained from in vitro-cultured biomass of different species of *Ruta*, such as *R. chalepensis*, *R. corsica*, or *R. graveolens*, showed a good bacteriostatic activity against *S. aureus* [22]. The results obtained in this study highlight that the biomass of *R. montana* produced in vitro by bioreactors can also represent a source of compounds with antibacterial activity, which are effective against resistant *S. aureus* strains as well.

### 2.5. Effect on Biofilm Formation

The *R. montana* extracts demonstrated a good capacity to reduce biofilm formation of methicillin-resistant *S. aureus* (MRSA) strains and biofilm producers (Figure 3). Compared to the demonstration on *S. aureus* 74CCH (53–27%) at 1/2 MIC, there was a higher activity on *S. aureus* 815 with a biofilm biomass reduction range from 78 to 37% and on *S. aureus* ATCC 43,300 with reduction from 71 to 28%. The extract 0.5./1.0 LS medium, particularly, reduced the biofilm formation of *S. aureus* 815 (78%) and *S. aureus* 43,300 (69%) followed by the 0.5/05 LS medium on *S. aureus* 43,300 (71%) and the 0.1/0.1 LS medium on *S. aureus* 815 (69%). The extract of the 0.5/0.5 LS medium showed a high activity against *S. aureus* ATCC 43,300 with a reduction in biofilm formation of 72% at 1/4 MIC. According to a study by Lemos et al. (2016), the action of the extracts was classified as highly effective in the range from ≤60% to ≤90% of the biofilm reduction. Statistically significant differences are indicated as * *p* < 0.05 vs. each control group [47]. The activity of extracts on strong biofilm producer strains is interesting. LS 0.5/1.0, followed by the LS 0.5/0.5 and 0.1/0.1 LS media variants among all the extracts, showed the best activity. Coumarins and alkaloids can interfere in biofilm production by repressing curli genes and motility genes, while catechins disrupt glycocalyx [48,49,50].

## 3. Materials and Methods

### 3.1. In Vitro Cultures

The starting material was the in vitro cultures of *R. montana*, established in 2018 from the seeds obtained from the Botanical Garden in Vácrátót, Hungary. The initial cultures were carried out in the form of liquid stationary cultures on Linsmaier and Skoog (LS) medium [51] and plant growth and development regulators: naphthyl-1-acetic acid (NAA) and 6-benzylaminopurine (BAP) at the concentration of 1.0/1.0 mg/L.

*Ruta montana* cultures were conducted in Plantform^TM^ bioreactors (Plant Form AB, Lomma, Sweden). For this, 15.0 g of previously grown plant biomass was placed in the bioreactors, which was then poured over with 500 mL of LS medium. Growth and development regulators in five concentration variants were added to it (Table 5). Immersion frequency was 5 min every 90 min. The cultures were maintained at 25 ± 2 °C under constant artificial light (16 µM·m^−2^·s^−2^). The cultivation was carried out for a period of 5 and 6 weeks. After this time, fresh biomass was collected and dried at approximately 38 °C.

### 3.2. Extraction

Briefly, 1.0 g of micronized dry biomass was weighed into 250 mL round bottom flasks and 50 mL of methanol was extracted from the flasks for 2 h at the solvent boiling point (64.7 °C). Following this, the extracts were evaporated. They were then dissolved in 4.0 mL high-performance liquid chromatography (HPLC)-grade methanol and filtered through Millipore membrane filters with a pore size of 0.22 µm for the HPLC analysis. Micronized dry biomass, 6–10 g of DW obtained after 5-week growth cycle, was used to prepare the extracts for the biological assays following the above-mentioned procedure.

### 3.3. Phytochemical Investigation

#### 3.3.1. Total Phenolic, Flavonoid, and Condensed Tannin Content

The spectrophotometric methods were used to determine the total phenolic, flavonoid, and condensed tannin content of *R. montana* extracts. The Folin–Ciocâlteu method was used to measure the total phenolic content with gallic acid used as a standard phenolic compound [52]. An aliquot of 0.1 mL of each sample solution was mixed with 0.2 mL Folin–Ciocâlteu reagent, 2 mL of distilled water, and 1 mL of 15% Na_2_CO_3_. A linear calibration curve of gallic acid in the range from 125 to 500 µg/mL was constructed. After a 2 h incubation at room temperature, the absorbance was measured at 765 nm, using a UV-1601 spectrophotometer (Shimadzu, Milan, Italy). The total phenolic content was expressed as mg gallic acid equivalents (GAE)/g of extract (dw) ± standard deviation (SD).

The aluminum chloride spectrophotometric assay was used to measure the total flavonoid content of the extracts [53]. Appropriately diluted 250 μL of each sample solution was mixed with 750 µL MeOH, 50 µL μL of 10% aluminum chloride, 50 µL of 1 M potassium acetate, and 1.4 mL of distilled water. The samples were incubated at room temperature in the dark for 30 min, and the reaction absorbance was measured at 415 nm using a UV-1601 spectrophotometer (Shimadzu, Milan, Italy). Quercetin was used to make the calibration curve, and the total flavonoid content was expressed as mg quercetin equivalents (QE)/g extract (dw) ± SD.

The vanillin method was used to determine the condensed tannin content of the extracts [54]. Briefly, 25 µL of each sample solution was mixed with 750 µL of 4% vanillin in MeOH and 375 µL of concentrated hydrochloric acid. After incubation at room temperature in the dark for 20 min, the absorbance of the reaction mixture was measured at 500 nm. The (+)-catechin was used as the reference standard, and the results were estimated as catechin equivalents (CE) and mg CE/g extract (dw) ± SD.

The results of the spectrophotometric determinations were obtained from the average of three independent experiments.

#### 3.3.2. HPLC Analyses

Reversed-phase HPLC analysis was performed as described elsewhere [55] on a Merck–Hitachi liquid chromatograph (LaChrom Elite, Hitachi, Tokyo, Japan) equipped with a DAD detector L-2455 and a Purospher ^®^ RP-18e 250 × 4 mm/5 mm column (Merck, Darmstadt, Germany). The analysis was conducted at 25 °C, with a mobile phase consisting of A—methanol, B—methanol:0.5% acetic acid 1:4 (*v/v*). The gradient elution, at the flow rate of 1 mL min^–1^, was as follows: 100% B for 0–20 min; 100–80% B for 20–35 min; 80–60% B for 35–55 min; 60–0% B for 55–70 min; 0% B for 70–75 min; 0–100% B for 75–80 min; 100% B for 80–90 min; wavelength range 200–400 nm. The quantification was performed at λ = 254 and 330 nm (phenolic acids, catechins, coumarins, and alkaloids) and at 330 and 370 nm (flavonoids).

The standards were purchased from the following companies: bergapten, imperatorin, xanthotoxin, and psoralen from Roth (Karlsruhe, Germany); caffeic acid, chlorogenic acid, cinnamic acid, ellagic acid, gallic acid, gentizic acid, isoferulic acid, neochlorogenic acid, *o*-coumaric acid, protocatechuic acid, rosmarinic acid, salicylic acid, sinapic acid, syringic acid, apigenin, apigetrin (apigenin 7-glucoside), hyperoside (quercetin 3-*O*-galactoside), isoquercetin (quercetin 3-*O*-glucoside), isorhamnetin, kaempferol, luteolin, myricetin, populnin (kaempferol 7-*O*-glucoside), robinin (kaempferol 3-*O*-robinoside-7-*O*-rhamnoside), quercetin, quercitrin (quercetin 3-*O*-rhamnoside), rhamnetin, rutoside, vitexin, 5,7-dimethoxycoumarin, 4-hydroxy-6-methylcoumarin, 6-methylcoumarin, osthole, and umbelliferone from Sigma-Aldrich (St Louis, MO, USA); *p*-coumaric acid, vanillic acid, ferulic acid, *p*-hydroxybenzoic acid, coumarin, and scopoletin from Fluka (Bucha, Switzerland); caftaric acid, cryptochlorogenic acid, isochlorogenic acid, catechin, epigallocatechin, epicatechin gallate, epicatechin, epigallocatechin gallate, cinaroside (luteolin 7-*O*-glucoside), osthenol, 4-methylumbelliferone, 4,6-dimethoxy-2H-1-benzopyran-2-one, and skimmianine from ChromaDex (Irvine, CA, USA); 4-O-feruloylquinic acid, apigetrin (apigenin 7-*O*-glucoside), apigenin 7-*O*-glucuronide, astragalin (kaempferol 3-*O*-glucoside), avicularin (quercetin 3-*O*-α-L-arabinofuranoside), trifolin (kaempferol 3-*O*-galactoside), isopimpinellin, isoimperatorin, daphnetin 7-methyl ether, rutaretin, daphnetin, osthenol, bergaptol, daphnetin dimethyl ether, γ-fagarine, and 7-isopentenyloxy-γ-fagarine from ChemFaces (Wuhan, China).

### 3.4. Antioxidant Activity

#### 3.4.1. DPPH Assay

The DPPH (1,1-diphenyl-2-picrylhydrazyl) assay was carried out to evaluate the free radical scavenging activity of *R. montana* extracts, using the protocol reported in a previous study by the authors [56]. The extracts were tested in the range from 0.0625 to 2 mg/mL, and butylated hydroxytoluene (BHT) was utilized as positive control. A volume of 0.5 mL of each sample was mixed with 3 mL of methanol DPPH solution (0.1 mM) and incubated in the dark for 20 min at room temperature. Then, the color change of the solutions was estimated by measuring the absorbance using a spectrophotometer (UV-1601, Shimadzu) at the wavelength of 517 nm. The scavenging activity was measured as the decrease in absorbance of the samples vs. DPPH control solution. The radical scavenging activity percentage (%) was calculated by the formula [(A_o_ − A_c_)/A_o_] × 100, where A_o_ is the absorbance of the control and A_c_ is the absorbance in the presence of the sample or standard. Three independent experiments were carried out, and the results are reported as the mean radical scavenging activity percentage (%) ± SD and mean 50% inhibitory concentration (IC_50_) ± SD defined as the concentration of the essential oil necessary to reduce or inhibit 50% of DPPH radical solution. The best activity against the DPPH radical was obtained with the lowest value of IC_50_. IC_50_ values were estimated by a nonlinear regression curve with the use of Prism Graphpad Prism version 9.0 for Windows, GraphPad Software, San Diego, CA, USA (www.graphpad.com (accessed on 10 January 2023)). The dose–response curve was obtained by plotting the percentage of inhibition *versus* the concentrations.

#### 3.4.2. Reducing Power Assay

The Fe^3+^–Fe^2+^ transformation method was used to estimate the reducing power of *R. montana* extracts [57]. The extracts were tested in the range from 0.0625 to 2 mg/mL, and BHT and ascorbic acid were utilized as positive controls. One mL of each sample was added after mixing 2.5 mL of phosphate buffer (0.2 M, pH 6.6) and 2.5 mL of 1% potassium ferricyanide. Following the incubation at 50 °C for 20 min and rapid cooling, 2.5 mL of 10% trichloroacetic acid was added, and the mixture was centrifuged (3000 rpm, 10 min). Then, 2.5 mL of the supernatant was transferred into another test tube and mixed with 2.5 mL of distilled water and 0.5 mL of 0.1% ferric chloride (FeCl_3_). After a 10-min incubation in the dark at room temperature, the color change of the samples was estimated by measuring absorbance at 700 nm. Three independent experiments were carried out, and the results are expressed as the mean absorbance values ± SD and ascorbic acid equivalents (ASE/mL) ± SD.

#### 3.4.3. Ferrous Ion (Fe^2+^) Chelating Activity Assay

The spectrophotometric measurement of the Fe^2+^-ferrozine complex was used following the protocol of Deker and Welch to determine the Fe^2+^ chelating activity of the *R. montana* extracts [58]. The extracts were tested in the range from 0.0625 to 2 mg/mL, and ethylenediaminetetraacetic acid (EDTA) was used as positive control. After mixing 1 mL of each sample with 0.5 mL of methanol and 0.05 mL of 2 mM FeCl_2_, 0.1 mL of 5 mM ferrozine was added to initiate the reaction. The mixture was incubated in the dark at room temperature for 10 min, and the color change of the solutions was estimated by measuring absorbance spectrophotometrically at 562 nm. Three independent experiments were carried out, and the results are reported as the mean inhibition of the ferrozine–(Fe^2+^) complex formation (%) ± SD and IC_50_ ± SD.

### 3.5. Antibacterial Bioassays

#### 3.5.1. Bacterial Strains

The antibacterial activity of *R. montana* extracts was tested against the following strains: *S. aureus* ATCC 6538, *S. aureus* ATCC 43300, *S. aureus* 815, *S. aureus* 74CCH, *S. epidermidis* ATCC 35984, *Escherichia coli* ATCC 25922, *E. coli* ATCC 10536, *E. coli* DSM 105388, *Pseudomonas aeruginosa* ATCC 9027, and *P. aeruginosa* DSM 102273 [59]. The strains were stored in the private collection of the Department of Chemical, Biological, Pharmaceutical and Environmental Sciences, University of Messina (Italy). They were stored at −70 °C in Microbanks™ (Pro-lab Diagnostics, Neston, UK). All reagents were purchased from Sigma-Aldrich (Milan, Italy), unless otherwise specified in the text.

#### 3.5.2. Antibacterial Screening

The minimum inhibitory concentration (MIC) and the minimum bactericidal concentration (MBC) of *R. montana* extracts were established according to the Clinical and Laboratory Standards Institute, with a few modifications [60]. Overnight cultures of the bacterial strains were grown at 37 °C in Mueller–Hinton Broth (MHB; Oxoid, Milan, Italy). The methanol extracts were dissolved in dimethyl sulfoxide (DMSO) and further diluted using MHB to obtain a final concentration of 2 mg/mL. Two-fold serial dilutions were prepared in a 96-well plate. The tested concentrations ranged from 1000 to 7.8 μg/mL. Working bacterial cultures were adjusted to the required inoculum of 1 × 10^5^ CFU/mL. Positive controls (medium with inocula, but without the extracts) and vehicle controls (medium with inocula and DMSO) were included. The concentration of solvent (DMSO) did not exceed 1%. Tetracycline was tested against all bacteria at concentrations ranging from 32 to 0.016 μg/mL. To determine MBC, aliquots (10 µL) were taken from each well and inoculated in Mueller–Hinton Agar (MHA, Oxoid, Basingstoke, UK). The cultures were incubated for 24 h at 37 °C. The bacterial growth was indicated visually and by a developer of the enzymatic activity (triphenyl tetrazolium chloride 0.05%), which reveals bacterial growth of purple color after 15 min of heating at 37 °C. MIC was defined as the lowest concentration of the extracts that completely inhibited growth compared to the growth controls. MBC was defined as the lowest concentration of extracts that did not allow visible growth when the aliquots of the well contents were plated on MHA and grown for 24 h at 37 °C. All experiments were repeated thrice in duplicate.

#### 3.5.3. Effect on Biofilm Formation

The effect of *R. montana* extracts on the biofilm formation by *Staphylococcus* strains was assayed based on the results obtained from the antibacterial screening. *Staphylococcus aureus* 815 and *S. aureus* 74CCH clinical isolates and *S. epidermidis* ATCC 35984 reference strain, among all the strains, are known as slime producers. They have been selected for their icaA/icaD gene presence, slime production, capability of forming biofilm on polystyrene surface, hemolytic activity, and agar typing [61]. The antibiofilm effect was assessed according to the description by Cramton et al., with a few modifications [54]. Overnight culture in 10 mL TSB with 1% glucose (TSBG) was diluted to standardize the *Staphylococcus* strain suspensions (1 × 10^6^ CFU mL). Aliquots of 100 µL were dispensed into each well of the sterile flat-bottom 96-well polystyrene microtiter plates (Corning Inc., Corning, NY) in the presence of 100 µL subinhibitory concentration (1/2 and 1/4 MIC) of each extract or 100 µL medium (control). The microtiter plates were incubated for 24 h at 37 °C. Positive biofilm controls (cells + TSBG) and negative controls (TSBG) were included. Planktonic growth was determined by spectrophotometric values (OD_492_ nm) using the microplate reader. The medium was then aspirated and the wells, rinsed twice with phosphate-buffered saline, were fixed by drying for 1 h. Once the wells were fully dry, 200 µL of 0.1% safranin was added for 2 min. The content of the wells was then aspirated, and 200 µL of 30% acetic acid (*v/v*) was added to the wells after they were rinsed with water, for the spectrophotometric analysis (OD_492_ nm). The results were derived from three separate experiments. The OD_492_ nm value obtained for each *Staphylococcus* strain without the extracts was used as the control. The ratio between the values of OD_492_ nm with and without the extracts was used to calculate the reduction percentage of biofilm formation in the presence of different extracts, adopting the following formula: [(OD_492_ nm with extract/OD _492_ nm without extract) × 100].

### 3.6. Statistical Analysis

Statistical comparison of the HPLC data was performed using the two-way analysis of variance (ANOVA), followed by the NIR post hoc test (STATISTICA v. 13.3 software, StatSoft, Inc., Tulsa, OK, USA). Three replicates were performed for each treatment. The comparison of the data obtained from the spectrophotometric determinations and antioxidant and antibiofilm tests was made using the ANOVA, followed by Tukey–Kramer multiple comparisons test (GraphPAD Prism Software for Science). *p*-values less than 0.05 were considered to be statistically significant. Detailed results of the statistical analysis are included in the Appendix A.

## 4. Conclusions

The study proved the ability of *R. montana* bioreactor cultures to accumulate secondary metabolites from various groups of compounds such as alkaloids, coumarins, flavonoids, and catechins. Furanocoumarins, xanthotoxin and bergapten in particular, and furoquinoline alkaloids were produced in the highest amounts. Thus, *R. montana* cultures can be proposed as a biotechnological source to obtain these valuable compounds. Further, *R. montana* bioreactor cultures showed good chelating properties as well as antibacterial and antibiofilm efficacy against resistant *Staphylococcus* strains. The use of a modern temporary immersion system (Plantform^TM^) allows for obtaining a large amount of biomass containing secondary metabolites with therapeutical potential in a short time.

## Figures and Tables

**Figure 1 ijms-24-07045-f001:**
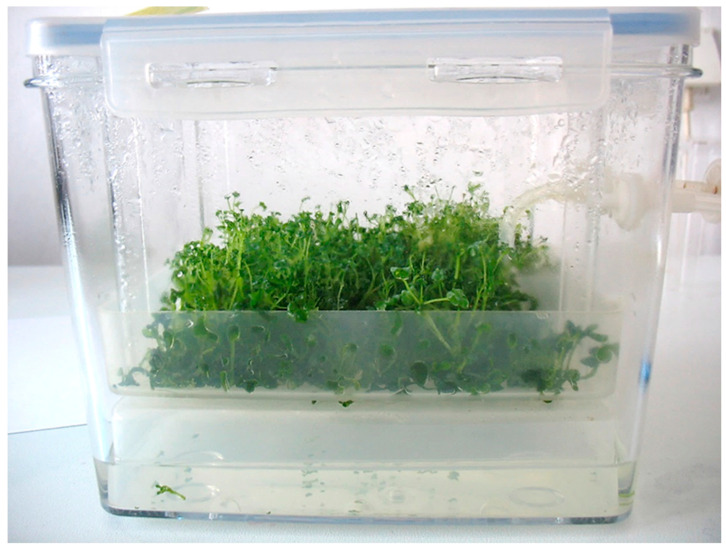
*Ruta montana* bioreactor culture (LS NAA/BAP 0.1/0.1 mg/L, 5-week growth cycle).

**Figure 2 ijms-24-07045-f002:**
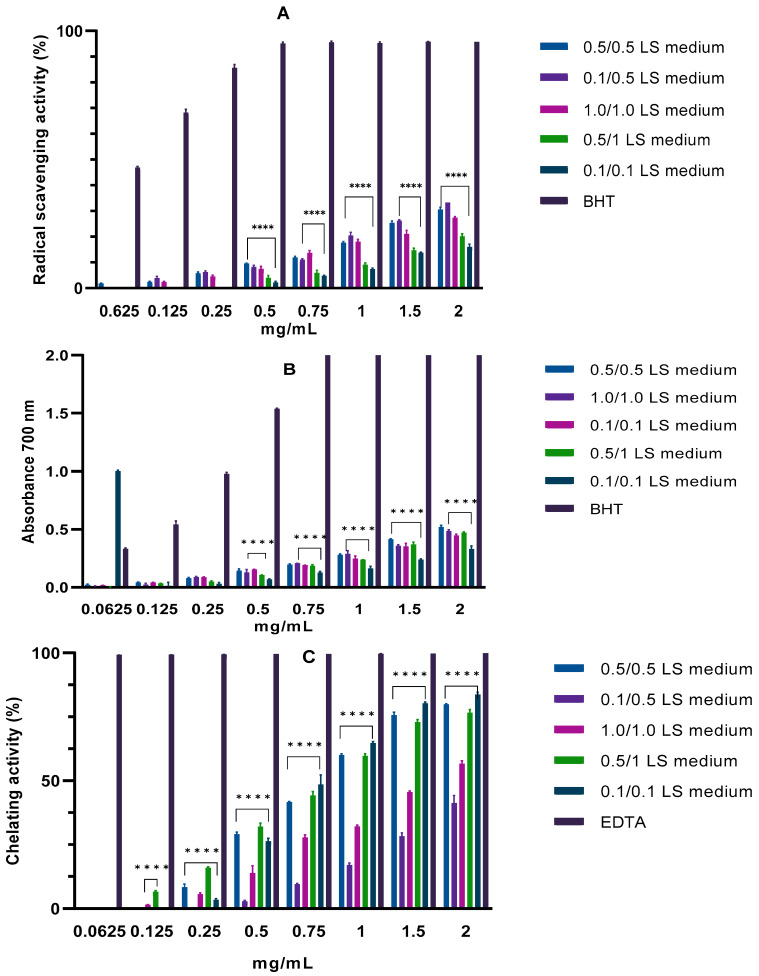
Free radical scavenging activity (DPPH assay) (**A**), reducing power (**B**), and ferrous ion chelating activity (**C**) of methanol extracts obtained from biomass of *R. montana* bioreactor cultures 2 grown on LS medium variant supplemented with different concentrations of NAA/BAP mg/L 293 (0.5/0.5, 0.1/0.5, 1.0/1.0, 0.1/0.1), after 5-week growth cycle. Reference standard: BHT (**A**,**B**), EDTA (**C**). Values are expressed as the mean ± SD (*n* = 3). Statistically significant differences between different variant are indicated as **** *p* < 0.0001.

**Figure 3 ijms-24-07045-f003:**
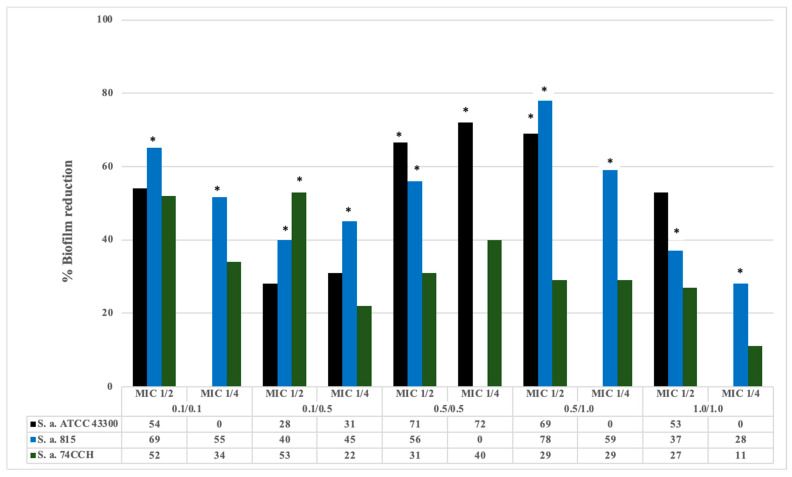
The effect of methanol extracts obtained from the biomass of *R. montana* bioreactor cultures grown on LS medium variant supplemented with different concentrations of NAA/BAP mg/L (0.1/0.1, 0.1/0.5, 0.5/0.5, 0.5/1.0, 1.0/1.0), 5-week growth cycle, on *S. aureus* strains biofilm formation reduction. The reduction percentage of biofilm formation was calculated using the following formula: [(OD_492_ nm with extract/OD _492_ nm without extract) × 100]. Statistically significant differences are indicated as * *p* < 0.05 vs. each control group.

**Table 1 ijms-24-07045-t001:** The dry weight (DW) [g] obtained from *R. montana* bioreactor cultures for different growth cycle and LS medium variants.

Growth Cycle	LS Medium Variant NAA/BAP (mg/L)				
	LS 0.1/0.1	LS 0.1/0.5	LS 0.5/0.5	LS 0.5/1.0	LS 1.0/1.0
5 weeks	7.734 ± 0.31 ^ab^	7.775 ± 0.36 ^ab^	8.030 ± 0.47 ^b^	8.042 ± 0.31 ^b^	7.738 ± 0.59 ^ab^
6 weeks	7.247 ± 0.86 ^a^	7.916 ± 0.02 ^ab^	7.265 ± 0.18 ^a^	7.787 ± 0.25 ^ab^	7.946 ± 0.21 ^ab^

Mean of three replications ± SD. Different letters (a, b) indicate significant differences (*p* < 0.05).

**Table 2 ijms-24-07045-t002:** Average content of metabolites [mg/100 g DW] in methanol extracts obtained from biomass of *R. montana* bioreactor cultures depending on the duration of the culture (5 and 6 weeks) growth cycle and LS medium variant.

	LS Medium Variant	Growth Cycle	
Accumulated Compounds	NAA/BAP (mg/L)	5 Weeks	6 Weeks
Bergapten	0.1/0.1	435.32 ± 50.26 ^bcd^	403.75 ± 35.98 ^abcd^
	0.1/0.5	375.70 ± 27.51 ^a^	364.99 ± 17.31 ^a^
	0.5/0.5	440.91 ± 21.31 ^cd^	395.18 ± 34.86 ^abcd^
	0.5/1.0	386.32 ± 30.63 ^ab^	378.85 ± 21.30 ^a^
	1.0/1.0	445.99 ± 18.08 ^d^	388.82 ± 42.73 ^abc^
Isoimperatorin	0.1/0.1	104.01 ± 9.72 ^de^	86.44 ± 3.95 ^c^
	0.1/0.5	79.11 ± 5.75 ^c^	56.35 ± 14.32 ^b^
	0.5/0.5	87.65 ± 7.76 ^cd^	35.23 ± 2.37 ^a^
	0.5/1.0	84.00 ± 9.72 ^c^	82.81 ± 18.34 ^c^
	1.0/1.0	105.78 ± 7.29 ^e^	82.33 ± 8.48 ^c^
Isopimpinellin	0.1/0.1	81.09 ± 8.77 ^a^	79.50 ± 7.52 ^a^
	0.1/0.5	100.75 ± 13.77 ^ab^	148.58 ± 6.99 ^c^
	0.5/0.5	92.89 ± 0.61 ^ab^	154.44 ± 20.64 ^c^
	0.5/1.0	90.25 ± 14.70 ^ab^	168.38 ± 12.19 ^c^
	1.0/1.0	113.97 ± 16.58 ^b^	223.21 ± 36.68 ^d^
Psoralen	0.1/0.1	340.07 ± 14.03 ^e^	236.59 ± 40.05 ^d^
	0.1/0.5	182.13 ± 15.72 ^bc^	185.22 ± 22.54 ^c^
	0.5/0.5	150.21 ± 11.24 ^a^	181.77 ± 9.57 ^bc^
	0.5/1.0	152.67 ± 6.37 ^ab^	169.00 ± 12.13 ^abc^
	1.0/1.0	168.45 ± 14.68 ^abc^	150.23 ± 1.91 ^a^
Xanthotoxin	0.1/0.1	863.78 ± 57.12 ^f^	772.92 ± 45.05 ^cde^
	0.1/0.5	682.89 ± 26.80 ^ab^	825.65 ± 64.82 ^def^
	0.5/0.5	885.92 ± 34.83 ^f^	846.06 ± 18.84 ^ef^
	0.5/1.0	731.96 ± 50.07 ^abc^	771.51 ± 47.02 ^cde^
	1.0/1.0	748.75 ± 38.92 ^bcd^	665.85 ± 57.15 ^a^
Total coumarins	0.1/0.1	1824.26 ± 98.07 ^d^	1579.20 ± 50.29 ^bc^
	0.1/0.5	1420.58 ± 35.81 ^a^	1580.80 ± 68.30 ^bc^
	0.5/0.5	1657.57 ± 45.79 ^c^	1612.67 ± 49.17 ^bc^
	0.5/1.0	1445.20 ± 86.25 ^ab^	1570.55 ± 18.55 ^bc^
	1.0/1.0	1582.94 ± 33.95 ^bc^	1510.44 ± 104.38 ^ab^
γ-Fagarine	0.1/0.1	305.42 ± 19.28 ^e^	215.82 ± 12.41 ^d^
	0.1/0.5	172.27 ± 36.64 ^c^	159.87 ± 15.20 ^bc^
	0.5/0.5	146.64 ± 6.45 ^abc^	151.05 ± 13.10 ^abc^
	0.5/1.0	133.85 ± 8.53 ^ab^	128.08 ± 18.70 ^a^
	1.0/1.0	133.61 ± 13.47 ^ab^	130.70 ± 13.75 ^ab^
Isopentenyloxy-γ-fagarine	0.1/0.1	30.79 ± 3.61 ^d^	39.42 ± 3.45 ^e^
	0.1/0.5	18.59 ± 2.40 ^a^	30.66 ± 1.57 ^cd^
	0.5/0.5	26.17 ± 2.02 ^bc^	42.20 ± 4.30 ^e^
	0.5/1.0	21.93 ± 1.64 ^ab^	32.88 ± 0.72 ^d^
	1.0/1.0	25.90 ± 2.26 ^b^	22.34 ± 3.02 ^ab^
Skimmianine	0.1/0.1	225.46 ± 14.16 ^fg^	126.58 ± 12.12 ^c^
	0.1/0.5	197.64 ± 18.26 ^ef^	89.38 ± 6.45 ^b^
	0.5/0.5	195.22 ± 23.06 ^ef^	51.95 ± 14.28 ^a^
	0.5/1.0	233.73 ± 27.88 ^g^	199.69 ± 30.19 ^ef^
	1.0/1.0	179.84 ± 24.17 ^de^	158.27 ± 4.47 ^cd^
Total alkaloids	0.1/0.1	561.66 ± 29.70 ^f^	381.82 ± 15.50 ^e^
	0.1/0.5	388.50 ± 25.29 ^e^	279.91 ± 22.69 ^ab^
	0.5/0.5	368.04 ± 31.01 ^de^	245.20 ± 2.97 ^a^
	0.5/1.0	389.51 ± 22.39 ^e^	360.66 ± 21.15 ^de^
	1.0/1.0	339.35 ± 17.92 ^cd^	311.31 ± 12.33 ^bc^
Catechin	0.1/0.1	89.62 ±0.97 ^f^	50.80 ±6.41 ^bc^
	0.1/0.5	76.12 ±11.16 ^e^	59.25 ±3.91 ^cd^
	0.5/0.5	59.70 ±5.98 ^d^	46.38 ±3.68 ^b^
	0.5/1.0	61.16 ±2.69 ^d^	22.01 ±1.83 ^a^
	1.0/1.0	73.56 ±3.51 ^e^	60.82 ±1.91 ^d^

Means of three measurements ± SD. Different letters ^(a–g)^ indicate significant differences between homogenous groups (*p* < 0.05) between LS medium variants for each compound after a certain growth cycle.

**Table 3 ijms-24-07045-t003:** Total phenolic, flavonoid, and condensed tannin content, free radical scavenging activity (DPPH) test, reducing power, and ferrous ion chelating activity of methanol extracts obtained from biomass of *R. montana* bioreactor cultures grown on LS medium variant supplemented with different concentrations of NAA/BAP (0.5/0.5, 0.1/0.5, 1.0/1.0, 0.1/0.1 mg/L), after 5-week growth cycle.

LS Medium Variant NAA/BAP (mg/L)	Total Polyphenols (mg GAE/g)	Total Flavonoids (mg QE/g)	Condensed Tannins (mg CE/g)	DPPH IC_50_ (mg/mL)	Reducing Power ASE/mL	Fe^2+^ Chelating Activity IC_50_ (mg/mL)
0.1/0.1	26.94 ± 0.67 ^a^	24.60 ± 0.58 ^b^	10.97 ± 0.5 ^c^	>2 mg/mL	28.95 ± 2.37 ^b^	0.90 ± 0.03 ^a^
0.1/0.5	41.61 ± 0.77 ^e^	45.65 ± 0.33 ^e^	9.42 ± 0.57 ^b^	>2 mg/mL	26.62 ± 3.25 ^b^	2.47 ± 0.01 ^c^
0.5/0.5	29.64 ± 0.76 ^b^	16.96 ± 0.01 ^a^	4.68 ± 0.32 ^a^	>2 mg/mL	19.54 ± 0.64 ^a^	0.94 a ± 0.02 ^a^
0.5/1.0	31.10 ± 0.88 ^c^	28.44 ± 0.20 ^c^	5.40 ± 0.2 ^a^	>2 mg/mL	39.75 ± 0.50 ^c^	0.93 ± 0.03 ^a^
1.0/1.0	34.32 ± 0.22 ^d^	33.00 ± 0.70 ^d^	6.21 ± 0.44 ^a^	>2 mg/mL	18.28 ± 4.30 ^a^	1.68 ± 0.02 ^b^
Reference standard				BHT 0.07 ± 0.01	BHT 1.44 ± 0.02 ^d^	EDTA 0.01 ± 0.001 ^d^

Values are expressed as mean ± SD (n = 3). ^a–e^ Different letters within the same column indicate significant differences between mean values (*p* < 0.05).

**Table 4 ijms-24-07045-t004:** MIC and MBC values (µg/mL) of methanol extracts obtained from biomass of *R. montana* bioreactor cultures grown on LS medium variant supplemented with different concentrations of NAA/BAP, 5-week growth cycle.

	LS Medium Variant NAA/BAP									
Strains	0.1./0.1		0.1/0.5		0.5/0.5		0.5/1.0		1.0/1.0	
	MIC	MBC	MIC	MBC	MIC	MBC	MIC	MBC	MIC	MBC
*S. aureus* ATCC 6538	125	500	1000	n.a	500	1000	n.a.	n.a.	1000	1000
*S. aureus* ATCC 43300	125	250	125	n.a	125	n.a.	n.a.	n.a.	500	1000
*S. aureus* 815	500	n.a.	1000	n.a	1000	n.a.	500	1000	n.a.	n.a.
*S. aureus* 74CCH	125	250	1000	n.a	500	n.a.	125	n.a.	500	n.a.
*S. epidermidis* ATCC 35984	500	500	500	n.a	250	n.a.	250	n.a.	1000	n.a.
*E. coli* ATCC 10536	500	1000	n.a.	n.a	n.a.	n.a.	n.a.	n.a.	n.a.	n.a.

n.a. not active at concentrations assayed.

**Table 5 ijms-24-07045-t005:** Concentration of plant growth regulator [mg/L] in LS medium variants.

Growth Regulator	LS Medium Variant NAA/BAP (mg/L)				
	0.1/0.1	0.1/0.5	0.5/0.5	0.5/1.0	1.0/1.0
NAA	0.1	0.1	0.5	0.5	1.0
BAP	0.1	0.5	0.5	1.0	1.0

## Data Availability

Not applicable.

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
