# Peer review of "Studies on the Accumulation of Secondary Metabolites and Evaluation of Biological Activity of In Vitro Cultures of Ruta montana L. in Temporary Immersion Bioreactors"

_ijms, 2023, doi:10.3390/ijms24087045_

Round 1

Reviewer 1 Report (Previous Reviewer 3)

The study of the accumulation of secondary metabolites in plant cultures is of considerable interest to researchers in various specialties.  The use of a unique bioreactor for in vitro cultivation Ruta montana, the study of the effect of various combinations of auxin and cytokinin on growth and the formation of secondary metabolites in them, allows optimizing and regulating the accumulation of secondary metabolites. Data on the assessment of the biological activity of extracts obtained from them are also interesting. All this allows us to recommend accepting the manuscript for publication.

Author Response

Dear Reviewer,

We are greatly obliged for having received the Reviewers’ valuable opinion. We are very grateful for your positive feedback regarding our article. We made corrections in accordance with the comments of other reviewers. All changes in the article have been marked in yellow.

Please, accept my best regards,

Yours sincerely,

Agnieszka Szewczyk

Reviewer 2 Report (Previous Reviewer 2)

1. This is my third time evaluating this manuscript and I found all the major issues including the problem of extracts-type research, statistics, and the unclear note for all tables still there. Basically, the authors can not fix the problems or understand them, therefore, I'm not willing to recommend it for publication.

2. L109, L184: It's still difficult to understand. How to compare them? Between variants or cycles?

3. Figure 2: The results of significant effects are missing.

4. Table 4: Very strange data, can not find a trend.

5. Figure 3: It is not possible to understand because of the poor layout.

6. The evaluation of the bioactivities of the extracts that contain alkaloids, coumarins, flavonoids, and catechins, is too preliminary, it was suggested to test more than the part of anti-Staphylococcus.

7. L13: Give the full scientific name.

8. L91: Don't understand what do you mean by "biomass growth".

9. L535-541: How many replicates were performed for each treatment?

Author Response

Dear IJMS Editor,

Dear Reviewer,

Thank you for the work you put into reviewing our article. We regret that our clarifications and corrections to previous reviews were not accepted by the reviewer. Below are detailed responses to the reviewer's suggestions:

“1. This is my third time evaluating this manuscript and I found all the major issues including the problem of extracts-type research, statistics, and the unclear note for all tables still there. Basically, the authors can not fix the problems or understand them, therefore, I'm not willing to recommend it for publication.”

Answer: We tried to apply all the corrections suggested by the reviewer. Below we remind of the reviewer's comments and our responses to them:

Responses to the Reviewer 2 comments (round1):

“In my opinion, this manuscript is out of the scope of this journal, because it doesn't present any results about molecular mechanisms in plant secondary metabolites for their metabolism, production, or bioactivities. The suggestion is to transfer it to more suitable journals, such as Plants or Agronomy, etc.”

Answer: We agree that some of our research may fall outside the general scope of the journal, but our submission is for a special issue on a slightly different topic from the general scope. We are applying for a special issue entitled "Health Properties of Plant Bioactive Compounds: Immune, Antioxidant and Metabolic Effects" and we believe that the subject matter of our article fits the scope of the special issue.

“The novelty of this study is not very good because in vitro culture using a bioreactor is a routine technology for many plants in the literature.”

Answer: Of course, bioreactors in general have long been used in biotechnology. There are many different types of bioreactors, which are mainly used in bacterial and fungal cultures. In the case of plant cultures, the situation is more complicated due to the wide variety of types of in vitro cultures. Older types of bioreactors can be used for cell cultures (cell suspensions or immobilized cells). This type of culture is preferred for e.g. performing biotransformation reactions. Some cell cultures are also used to produce secondary metabolites. Unfortunately, most cell cultures, as cultures with the lowest organization, are not good material for the production of metabolites. The content of active compounds is usually very low, the metabolism leads to simple compounds, e.g. phenolic acids, the subsequent stages of the metabolic pathway are impaired. Hence the idea of cultivating cultures with a high degree of organization - e.g. shoot cultures, whose metabolism is more similar to that of the parent plants. Unfortunately, older types of bioreactors are completely unsuitable for shoot cultures. It was only in the last 10 years that a new technology appeared – temporary immersion, which made it possible to conduct shoot cultures on a large scale. Plantform bioreactors were first used for micropropagation. After finding their high efficiency in this area and confirming the high increase in biomass, there was an interest in cultivating this type of culture for the production of secondary metabolites (first reports since 2017).

“Introduction: Add a brief history of the use of bioreactors for in vitro culture of medicinal plants and secondary metabolite production.”

Answer: We have made the suggested corrections in the Introduction section.

“Evaluation of bioactivities: It's not acceptable for the testing of bioactivities of natural compounds using plant extracts because it always can not know which single compounds exert the main effects. Consequently, the authors never know the in-depth molecular mechanisms of the bioactivities.”

Answer: Thank you for this observation. The problem of research in the field of phytotherapy is very complicated. On the one hand, it would be most beneficial to isolate individual metabolites and study their biological activity. On the other hand, most natural plant materials used in herbal medicine are either specific raw materials, such as herbs, leaves, flowers, etc., or standardized plant extracts. Plant extracts are treated as a whole as so-called phytocomplexes, where individual compounds have a synergistic effect. Another positive aspect of using extracts that are a mixture of various metabolites is the reduction of side effects or toxicity characteristic of one of the components. When it comes to antioxidant activity, mainly plant extracts are tested, due to the synergistic effect of the entire complex of metabolites.

“The overall quality and the data are more fit to the journal "Molecules".”

Answer: Thank you for this suggestion. It is also a very valuable journal in which we plan to publish in the future.

Responses to the Reviewer 2 comments (round2):

“In my opinion, this manuscript should be transferred to a more suitable journal such as Molecules or Plants. Because it used plant extracts but not single compounds for study and it's not possible to explore any molecular mechanisms of bioactivities.”

Answer: We've included the explanations in the round one answer.

Also, in R1, I can say that all statistics look problematic or unclear, for example, in table 1, the significant differences between means are all wrong, and the letters only have "ab", "abc", but don't have "a". The authors did not mention how to compare different letters. Between variants or cycles? The same problem also happened in tables 2 and 3. The data of this study is unreliable.

Answer: We do not agree with this opinion. We also consulted our statistician and he confirmed that the statistical analysis of the results is correct. It should be taken into account that for each metabolite, as many as 10 results are compared with each other - in such a situation, it is rare for each result to differ significantly from each other. Of course, we understand that such a large number of results is a bit confusing for the reader. Therefore, we changed the method and we compare homogeneous groups only. Thus, the number of letters in the description has been reduced (a two-way ANOVA was performed, the dependent variable was the content of compounds, the independent variables were the growth cycle and medium variant). As suggested by the reviewer, we included a more detailed description of the statistical method in the text.

We emphasize that in the first round the reviewer did not mention anything about the statistical methods, only in the second round did enter his comments without giving us a chance to revise the manuscript earlier.

 In the current round, we have made new corrections marked in yellow in the text to describe the statistical methods more clearly. Statistics results are also included in Supplementary Materials. We hope that the corrections made will improve the readability of the manuscript.

  1. L109, L184: It's still difficult to understand. How to compare them? Between variants or cycles?

Answer: A two-way ANOVA was performed, the dependent variable was the content of compounds, the independent variables were the growth cycle and medium variant. Since there is a correlation between the medium variant and the growth cycle, we cannot compare these values separately in a one-way ANOVA. The results are given as significant differences between homogenous groups (p < 0.05) between LS medium variants for each compound after a certain growth cycle.

  1. Figure 2: The results of significant effects are missing.

Answer: The graphs have been redone reporting the significance

  1. Table 4: Very strange data, can not find a trend.

Answer: We changed the title of the table and eliminated the unit of measure for phytohormone concentration because it could confuse the reader.

  1. Figure 3: It is not possible to understand because of the poor layout.

Answer: We changed the graphic in order to clarify the results to the reader 

  1. The evaluation of the bioactivities of the extracts that contain alkaloids, coumarins, flavonoids, and catechins, is too preliminary, it was suggested to test more than the part of anti-Staphylococcus.

Answer: In this study we performed a screening of antibacterial activity of the R. montana extracts against (n. 5) Staphylococcus strains, (n. 3) Escherichia coli strains, (n. 2) Pseudomonas aeruginosa strains. The results showed the best activity of the extracts on the Staphylococcus strains. Among these we have chosen the resistant S. aureus strains with the lowest MIC value to evaluate the anti-biofilm activity of the extracts. This is the reason why we have not considered at this time to evaluate the activity of other species.

  1. L13: Give the full scientific name.

Answer: We do not know which word this remark refers to. Name of the species or type of bioreactor? If the name of the species, we added the full name "Ruta montana L.", if the name of the bioreactor, we added “temporary immersion”.

  1. L91: Don't understand what do you mean by "biomass growth".

Answer: It is a term commonly used in the terminology of plant biotechnology to describe the weight gain of e.g. shoots with callus tissue, obtained in cultures.

  1. L535-541: How many replicates were performed for each treatment?

Answer: For each treatment three replicates were performed.

Please, accept my best regards,

Yours sincerely,

Agnieszka Szewczyk

Reviewer 3 Report (New Reviewer)

In order to be published in IJMS, I consider that the manuscript needs several improvements. All detailed comments and suggestions can be found in the attached file.

Author Response

Dear Reviewer,

We are greatly obliged for having received the Reviewers’ valuable opinion and helpful suggestion on our manuscript. We made corrections in accordance with the comments. All changes in the article have been highlighted with yellow. The replies to the specific comments are listed below.

“Considering the instructions for authors of IJMS, the abstract should be a total of about 200
words maximum. As such, I recommend you to summarize the information from the abstract
in order to reduce its original (278 words) size.”

Answer: The abstract was reduced

“The text from line 38-40 has no reference, or is from the same source [1-4]?”

Answer: Thank you for this observation. We have made corrections.

“Why some parts of the manuscripts are highlighted with yellow?”

Answer: Since we were correcting the article, we marked the corrections in yellow. We have removed the previous highlighted parts of the manuscript. Now only current corrections are highlighted with yellow.

“Considering “the rich chemical composition of the species”, I recommend you to make a
Table containing the centesimal composition for those compounds that you consider to be
more important.”

Answer: Thank you for this observation. The phrase "rich chemical composition" is used incorrectly. We changed it to the phrase "diverse chemical composition". The quantitative chemical composition of R. montana is not yet sufficiently known, most of the literature reports concern the qualitative composition.

“Reformulate “Plant in vitro cultures” from lines 71-72 as in the line 69 (“In vitro plant
cultures”).”

Answer: The correction was done.

“I recommend you to make a short addition to the aims of the work (“Therefore, the present
study aimed to investigate the biosynthetic, antioxidant, antibacterial, and antibiofilm
potential of the in vitro cultures of R. montana” using a temporary immersion system
(PlantformTM). Subsequent, move the text from lines 74-77 (associated to the aim of the
work) at the end of the introduction section.”

Answer: The correction was done.

“I consider that some changes in Table 1 are required.
- The (proposed) title: The dry weight (DW) [g] obtained from R. montana bioreactor
cultures for different growth cycle and LS medium variants.
- Footer of the table: Mean of three replications ± SD. Different letters (a,b) indicate
significant differences (p < 0.05).
- Remove the text „a-b Letters represent homogeneous groups” from original footer.”

Answer: The title and the footer of the Table 1 were replaced.

“Reformulate the text from lines 98-99 as follows: There were no significant differences in
biomass growth after 5 and 6 weeks of the culture cycle for most of the LS medium. Only in
case of LS 0.5/0.5 medium variant, the dry biomass after 5 weeks was significantly higher
compared to those for 6 weeks (explain this!).”

Answer: The correction was done.

“Remove the text “Currently, there are no studies on R. montana in vitro cultures” from line
111, because it is already presented in lines 75-76.”

Answer: We removed the sentence from the text.

“Remove from the text different citations with authors name (Ekiert H. et al., 2001, Szewczyk
A et al., 2022).”

Answer: We removed the authors name from the text.

“The conclusions from lines 127-129 (“The best medium turned out to be the LS medium
containing 0.5/1.0 mg/L NAA/BAP. The most favorable increase in biomass occurred during
the 5-week growth cycle.”) have no statistical basis, given that you don’t made a statistical
comparison between the values of different LS mediums. Also, the values of average dry
weight for the 5 weeks (7.864 g) and 6 weeks (7.638 g) growth cycle, are to close to be
significant different. Therefore, if you found significant differences you have to prove them,
when not you have to reformulate these conclusions.”

Answer: Thank you for this suggestion. True, based on statistical analysis, it is impossible to indicate the best medium. Our wording was more about macroscopic observations of in vitro culture tissues. We changed this phrase to "There are no statistically significant differences between the media used."

“As in the case of Table 1, some changes in Table 2 are required.
- In Tables 1, 2, use the same expression: “growth cycle” or “growth period”, not both.
- The (proposed) title: Average content of metabolites [mg/100 g DW] in methanol
extracts obtained from biomass of R. montana bioreactor cultures depending on the duration
of the culture (5 and 6 weeks) growth cycle and LS medium variant.
- Footer of the table: Means of three measurements ± SD. Different letters (a,b,c,d)
indicate significant differences between homogenous groups (p < 0.05) between LS medium
variants for each compound after a certain growth cycle.
- Remove the text „The dependent variable was the content of compounds, the
independent variables were the growth cycle and medium variant” from original title.
- Remove the text „a-b Letters represent homogeneous groups” from original footer.”

Answer: The title and the footer of the Table 2 were replaced.

“Recommended changes for Table 3:
- The (proposed) title: Total phenolic, flavonoid, and condensed tannin content, free
radical scavenging activity (DPPH) test, reducing power, and ferrous ion chelating activity of
methanol extracts obtained from biomass of R. montana bioreactor cultures grown on LS
medium variant supplemented with different concentrations of NAA/BAP (0.5/0.5, 0.1/0.5,
1.0/1.0, 0.1/0.1 mg/L), after 5-week growth cycle.
- Footer of the table: Values are expressed as mean ± SD (n = 3). a–e Different letters
within the same column indicate significant differences between mean values (p < 0.05).”

Answer: The title and the footer of the Table 3 were replaced.

“Proposed title for Figure 2. Free radical scavenging activity (DPPH assay) (A), reducing
power (B), and ferrous ion chelating activity (C) of methanol extracts obtained from biomass
of R. montana bioreactor cultures 2 grown on LS medium variant supplemented with different
concentrations of NAA/BAP mg/L 293 (0.5/0.5, 0.1/0.5, 1.0/1.0, 0.1/0.1), after 5-week
growth cycle”

Answer: The title of the Figure 2 was changed.

“To denote significant differences using letters, as a rule it starts with a for the highest mean,
followed by b for the next mean, and so on. Given that in Table 2 you started with a for the
lowest mean (which is also correct), for a better understanding of the results (by the readers) I
recommend you to use the same system for significance letters in Table 3 (where you started
from top to the bottom of the column).”

Answer: We changed the letters in the Table 3 starting with “a” for the lowest mean.

“Remove “mg/L (0.5/0.5, 0.1/0.5, 1/1, 0.1/0.1)” from the title of Table 4.”

Answer: we removed “mg/L (0.5/0.5, 0.1/0.5, 1/1, 0.1/0.1)” from the title of Table 4.

“Given that all the experiments regarding antibacterial screening „were repeated thrice in
duplicate”, it’s possible to calculate SD for different MIC and MBC values and to make a
statistical comparison (using a multiple comparisons test) between LS medium variants. In
this way you can increase the scientific value of your results.”

Answer: It is not possible to calculate the standard deviations of MICs because the results are given by two replicas of the same dilution (2 fold dilution) of sample tested repeated thrice. So, the result obtained depends on the concentration of the diluted sample that has activity. Minimal Bactericidal Concentration (MBC) results depend on whether or not bacteria growth on solid medium after to be exposed to MIC concentration or to higher concentrations of the extracts tested. Standard deviation cannot be calculated.

“Explain in the M&M section how the inhibitory concentration IC50 (Method, Software) and
DPPH radical scavenging activity (Formulas) were calculated.”

Answer: We added the information required.

“In the M&M section there are no information regarding the in vitro culture conditions
relating to light (photoperiod, intensity) and temperature. These conditions should be
described with sufficient detail to allow others to replicate your experiments.”

Answer: We added the information required.

“For a better highlighting of the obtained results and considering that the data were processed
using ANOVA, we consider that it is (absolutely) necessary to present the results of ANOVA
in the manuscript or in the supplementary files.”

Answer: We added the information in Supplementary Materials.

“Considering that the Discussion should explain the findings and places the results in the context of other studies, you have to use more references (in the text there are only 3 references for subsections 2.1, 2.2.1, 2 references for subsection 2.2.2) to compare your results.”

Answer: Since there are no reports of in vitro cultures of R. montana, we compared our results with cultures of other rue species. As suggested, we have added more references to the discussion.

Please, accept my best regards,

Yours sincerely,

Agnieszka Szewczyk

Round 2

Reviewer 3 Report (New Reviewer)

The authors did a great work! Congratulations!

Author Response

Dear Reviewer,

We are greatly obliged for having received the Reviewers’ valuable opinion. We are very grateful for your positive feedback regarding our article.

Please, accept my best regards,

Yours sincerely,

Agnieszka Szewczyk

This manuscript is a resubmission of an earlier submission. The following is a list of the peer review reports and author responses from that submission.

Round 1

Reviewer 1 Report

In line 32 “MIC 125 g/mL” there is a lot of space between words

In table 1. "LS medium variant" the units in parentheses and NAA/BAP (mg/mL) are missing, the units in table 2, 5 are the same. Review all the tables and homogenize the units

Line 143, because 885.9 is in bold, the same in line 151, 156, 160, 163, and in table 2

The graphs in figures 2, 3 can be done with another program, to have a better graph and probably not use Excel, which seems to be the one they used. I do not understand why not use the statistical program GraphPAD to graph, the program gives excellent graphs.

In line 234, why the space? I was missing a letter from "-fagarine" and the same in line 250 in ( g/mL)

In table 4, n.a. put meaning in table footer

In line 256 "1/2" is not the same as in line 262 with "1/4" homogenize. It is probably corrected by inserting a symbol in the "Word" program

Author Response

Dear Reviewer,

We are greatly obliged for having received the Reviewers’ valuable opinion and helpful suggestion on our manuscript. All changes in the manuscript are marked in yellow. The replies to the specific comments are listed below.

“In line 32 “MIC 125 g/mL” there is a lot of space between words”

Answer: we put the symbol between the words

“In table 1. "LS medium variant" the units in parentheses and NAA/BAP (mg/mL) are missing, the units in table 2, 5 are the same. Review all the tables and homogenize the units”

Answer: we added the missing word and homogenized all the tables

“Line 143, because 885.9 is in bold, the same in line 151, 156, 160, 163, and in table 2”

Answer: We have made suggested corrections.

“The graphs in figures 2, 3 can be done with another program, to have a better graph and probably not use Excel, which seems to be the one they used. I do not understand why not use the statistical program GraphPAD to graph, the program gives excellent graphs.”

Answer:  We performed all the graphs with GraphPAD and we changed them in the manuscript

“In line 234, why the space? I was missing a letter from "-fagarine" and the same in line 250 in ( g/mL)”

Answer: Thank you. We inserted the appropriate symbol.

“In table 4, n.a. put meaning in table footer”

Answer: We put the meaning of n.a. in table footer.

"In line 256 "1/2" is not the same as in line 262 with "1/4" homogenize. It is probably corrected by inserting a symbol in the "Word" program"

Answer: Thank you. We corrected the words

Please, accept my best regards,

Yours sincerely,

Agnieszka Szewczyk

Reviewer 2 Report

1. In my opinion, this manuscript is out of the scope of this journal, because it doesn't present any results about molecular mechanisms in plant secondary metabolites for their metabolism, production, or bioactivities. The suggestion is to transfer it to more suitable journals, such as Plants or Agronomy, etc.

2. The novelty of this study is not very good because in vitro culture using a bioreactor is a routine technology for many plants in the literature.

3. Introduction: Add a brief history of the use of bioreactors for in vitro culture of medicinal plants and secondary metabolite production.

4. Evaluation of bioactivities: It's not acceptable for the testing of bioactivities of natural compounds using plant extracts because it always can not know which single compounds exert the main effects. Consequently, the authors never know the in-depth molecular mechanisms of the bioactivities.

5. The overall quality and the data are more fit to the journal "Molecules".

Author Response

Dear Reviewer,

We are greatly obliged for having received the Reviewers’ valuable opinion and helpful suggestion on our manuscript. All changes in the manuscript are marked in yellow. The replies to the specific comments are listed below.

“In my opinion, this manuscript is out of the scope of this journal, because it doesn't present any results about molecular mechanisms in plant secondary metabolites for their metabolism, production, or bioactivities. The suggestion is to transfer it to more suitable journals, such as Plants or Agronomy, etc.”

Answer: We agree that some of our research may fall outside the general scope of the journal, but our submission is for a special issue on a slightly different topic from the general scope. We are applying for a special issue entitled "Health Properties of Plant Bioactive Compounds: Immune, Antioxidant and Metabolic Effects" and we believe that the subject matter of our article fits the scope of the special issue.

“The novelty of this study is not very good because in vitro culture using a bioreactor is a routine technology for many plants in the literature.”

Answer: Of course, bioreactors in general have long been used in biotechnology. There are many different types of bioreactors, which are mainly used in bacterial and fungal cultures. In the case of plant cultures, the situation is more complicated due to the wide variety of types of in vitro cultures. Older types of bioreactors can be used for cell cultures (cell suspensions or immobilized cells). This type of culture is preferred for e.g. performing biotransformation reactions. Some cell cultures are also used to produce secondary metabolites. Unfortunately, most cell cultures, as cultures with the lowest organization, are not good material for the production of metabolites. The content of active compounds is usually very low, the metabolism leads to simple compounds, e.g. phenolic acids, the subsequent stages of the metabolic pathway are impaired. Hence the idea of cultivating cultures with a high degree of organization - e.g. shoot cultures, whose metabolism is more similar to that of the parent plants. Unfortunately, older types of bioreactors are completely unsuitable for shoot cultures. It was only in the last 10 years that a new technology appeared – temporary immersion, which made it possible to conduct shoot cultures on a large scale. Plantform bioreactors were first used for micropropagation. After finding their high efficiency in this area and confirming the high increase in biomass, there was an interest in cultivating this type of culture for the production of secondary metabolites (first reports since 2017).

“Introduction: Add a brief history of the use of bioreactors for in vitro culture of medicinal plants and secondary metabolite production.”

Answer: We have made the suggested corrections in the Introduction section.

“Evaluation of bioactivities: It's not acceptable for the testing of bioactivities of natural compounds using plant extracts because it always can not know which single compounds exert the main effects. Consequently, the authors never know the in-depth molecular mechanisms of the bioactivities.”

Answer: Thank you for this observation. The problem of research in the field of phytotherapy is very complicated. On the one hand, it would be most beneficial to isolate individual metabolites and study their biological activity. On the other hand, most natural plant materials used in herbal medicine are either specific raw materials, such as herbs, leaves, flowers, etc., or standardized plant extracts. Plant extracts are treated as a whole as so-called phytocomplexes, where individual compounds have a synergistic effect. Another positive aspect of using extracts that are a mixture of various metabolites is the reduction of side effects or toxicity characteristic of one of the components. When it comes to antioxidant activity, mainly plant extracts are tested, due to the synergistic effect of the entire complex of metabolites.

“The overall quality and the data are more fit to the journal "Molecules".”

Answer: Thank you for this suggestion. It is also a very valuable journal in which we plan to publish in the future.

Please, accept my best regards,

Yours sincerely,

Agnieszka Szewczyk

Reviewer 3 Report

The accumulation and composition of some secondary metabolites of rue culture microshoots grown on media with different balances of auxins and cytokinins were studied. The biological activity of the extracts obtained from them was traced.  The material is original and interesting for the reader.

The main remarks are noted in the manuscript of the article.

1. You should pay attention to the introduction section and note the importance of working with plant cultures grown in vitro.

2. What determines the choice of plant age (5 and 6 week)

3. No conclusions for each section of experimental data.

4. It is not necessary to repeat in the text of the manuscript the data that is already in the tables. You can note patterns and changes in these parameters under certain conditions.

5. Improve the presentation of data in tables. Think of media variant names that, in this form, are bad for the reader.

Author Response

Dear Reviewer,

We are greatly obliged for having received the Reviewers’ valuable opinion and helpful suggestion on our manuscript. All changes in the manuscript are marked in yellow. The replies to the specific comments are listed below.

“You should pay attention to the introduction section and note the importance of working with plant cultures grown in vitro.”

Answer: The Introduction section has been revised as suggested.

“What determines the choice of plant age (5 and 6 week)”

Answer: The 5th and 6th weeks listed in the publication are not actually the age of the plants. This term refers to the growth cycle (i.e. the time from the transfer of a small amount of culture biomass to fresh medium until the end of the experiment and collection of the grown biomass). R. montana cultures were established in 2018 and have been continuously maintained since then, so the age of the culture itself is 3 years old. The cultures are passaged about every 6 weeks on fresh media, because after a long time the nutrients in the medium are depleted and the cultures die. Therefore, it is necessary to constantly pass cultures. In vitro cultures are characterized by several growth phases during the growth cycle. At the beginning of the cycle, there is a lag phase, next an intensive growth phase, then the cultures move to a slow growth phase (stationary phase). The greatest accumulation of secondary metabolites occurs in the stationary phase. Our previous unpublished preliminary research showed that the stationary phase in our cultures falls on weeks 5 and 6 of the growth cycle.

“No conclusions for each section of experimental data.”

Answer: We have made the suggested corrections in each section of experimental data.

“It is not necessary to repeat in the text of the manuscript the data that is already in the tables. You can note patterns and changes in these parameters under certain conditions.”

Answer: We have made suggested corrections.

“Improve the presentation of data in tables. Think of media variant names that, in this form, are bad for the reader.

Answer: Thank you for this suggestion. The form of media version names was changed.

Please, accept my best regards,

Yours sincerely,

Agnieszka Szewczyk

Round 2

Reviewer 2 Report

In my opinion, this manuscript should be transferred to a more suitable journal such as Molecules or Plants. Because it used plant extracts but not single compounds for study and it's not possible to explore any molecular mechanisms of bioactivities. Also, in R1, I can say that all statistics look problematic or unclear, for example, in table 1, the significant differences between means are all wrong, and the letters only have "ab", "abc", but don't have "a". The authors did not mention how to compare different letters. Between variants or cycles? The same problem also happened in tables 2 and 3. The data of this study is unreliable.